# Snail Based Carbonated-Hydroxyapatite Material as Adsorbents for Water Iron (II)

**DOI:** 10.3390/ma15093253

**Published:** 2022-04-30

**Authors:** Bernard Owusu Asimeng, Edward Kwame Amenyaglo, David Dodoo-Arhin, Johnson Kwame Efavi, Bright Kwakye-Awuah, Elvis Kwason Tiburu, E. Johan Foster, Jan Czernuska

**Affiliations:** 1Department of Biomedical Engineering, University of Ghana, Legon, Accra LG 77, Ghana; ekamenyaglo@st.ug.edu.gh (E.K.A.); etiburu@ug.edu.gh (E.K.T.); 2Department of Materials Science and Engineering, University of Ghana, Legon, Accra LG 77, Ghana; ddodoo-arhin@ug.edu.gh (D.D.-A.); jkefavi@ug.edu.gh (J.K.E.); 3Department of Physics, Kwame Nkrumah University of Science and Technology, UPO, PMB, Kumasi AK 448, Ghana; bkwakye-awuah.cos@knust.edu.gh; 4Department of Chemical and Biological Engineering, University of British Columbia, Vancouver, BC V6T 1Z3, Canada; johan.foster@ubc.ca; 5Department of Materials, University of Oxford, Oxford OX1 3PH, UK; jan.czernuszka@materials.ox.ac.uk

**Keywords:** carbonated-hydroxyapatite, iron(II) adsorption, adsorption kinetics, water remediation

## Abstract

Carbonated hydroxyapatite (CHAp) adsorbent material was prepared from *Achatina achatina* snail shells and phosphate-containing solution using a wet chemical deposition method. The CHAp adsorbent material was investigated to adsorb aqua Fe(II) complex; [Fe(H_2_O)_6_]^2+^ from simulated iron contaminated water for potential iron remediation application. The CHAp was characterized before and after adsorption using infrared (IR) and Raman spectroscopy. The IR and the Raman data revealed that the carbonate functional groups of the CHAp adsorbent material through asymmetric orientation in water bonded strongly to the aqua Fe(II) complex adsorbate. The adsorption behaviour of the adsorbate onto the CHAp adsorbent correlated well to pseudo-second-order kinetics model, non-linear Langmuir and Freundlich model at room temperature of a concentration (20–100 mg L^−1^) and contact time of 180 min. The Langmuir model estimated the maximum adsorption capacity to be 45.87 mg g^−1^ whereas Freundlich model indicated an S-type isotherm curvature which supported the spectroscopy revelation.

## 1. Introduction

Pollutants such as iron, magnesium, and heavy metals including lead and mercury have increased in our water bodies [1,2]. For instance, iron is mostly present in drinking water as a result of the use of iron coagulants or corrosion of steel and cast-iron pipes during water distribution to homes, schools, hospitals, and workplaces [2]. Although, iron plays a vital role in the metabolic process including electron transport, oxygen transport and deoxyribonucleic acid (DNA) synthesis in living organisms [3], excessive levels above World Health Organization (WHO) permissible limits in drinking water (0.3 mg L^−1^) result in tissue damage and other health complications [1,2,3,4,5,6,7]. In addition, excessive iron levels (over 40 mg L^−1^) in water produce a red colour, a bad odour, and an unpleasant taste due to changes in pH, which makes it unpalatable [3,5,6].

Many treatment methods and processes including adsorption, solidity and ion exchange [7,8,9] have been employed over the years to remove or reduce the concentration of iron in drinking water by individuals and water treatment companies [10]. These methods have a relatively low level of adsorption efficiency [11]. More recently tried methods include nanomaterials such as silver, gold, graphene and carbon that have been incorporated to form nonporous membranes. The problem with the incorporation of the nanomaterials is that the nanomaterials disperse in water and excessive levels of the nanomaterials in water cause toxicity [11,12]. Thus, researchers are investigating more biocompatible materials, such as hydroxyapatite [HAp: Ca_10_(PO_4_)_6_OH_2_], which do not cause toxicity. However, the HAp has a low adsorption capacity for iron (II) [1,10]. Thus, the authors have substituted carbonate into the HAp [CHAp: Ca_10−x/2_(PO_4_)_6_CO_3−x_OH_2_] and we report on the preparation conditions and higher adsorption capacity of CHAp on iron (II). The CHAp was prepared from snail shells (*Achatina achatina*) and phosphate-containing solution. The CHAp before and after iron entrapment were characterized using infrared (IR) and Raman spectroscopy. The adsorption behaviour was studied using Pseudo-first and second-order kinetics, and nonlinear Langmuir and Freundlich model.

## 2. Materials and Methods

### 2.1. Materials

Calcite and Ammonium phosphate dibasic (APD) were used for the preparation of the CHAp adsorbent. Calcite was prepared from *Achatina achatina* snail shells collected from food vendors from Madina market in Ghana according to Asimeng et al. [13] method at 700 °C whereas APD (pH~8.12) purchased from Kosdag Listed Company. Ammonium ferrous sulphate (AFS: FeH_8_N_2_O_8_S_2_) was purchased from ACE Chemicals to simulate aqua complex of [Fe(H_2_O)_6_]^2+^ solution to represent Fe(II).

### 2.2. CHAp Adsorbent Preparation

Carbonated hydroxyapatite (CHAp) adsorbent was prepared from 0.3 M solution of APD, and calcite solution of 5.7 g of calcite mixed with 0.15 cm^3^ of distilled water [14]. The APD solution was added in dropwise to the calcite solution and stirred for 2 h using a magnetic stirrer. The mixture aged for 12 h and then filtered, dried for 4 h at 100 °C to obtain the CHAp adsorbent material.

### 2.3. Adsorption Experiments

Concentrations of 20–100 mg L^−1^ in steps of 20 mg L^−1^ of aqua Fe(II) complex; [Fe(H_2_O)_6_]^2+^ adsorbate were prepared from a standard solution of AFS. A volume of 1000 µL of each concentration was pipetted into 6 mg of CHAp in eppendorf tubes. The CHAp adsorbent material and the [Fe(H_2_O)_6_]^2+^ adsorbates were rotated for 3 h at intervals of 30 min at room temperature to allow for contact. Room temperature was selected for the experiment because HAp and HAp based materials are most stable [15,16]. The mixture was centrifuged at 6000× *g* rpm for 10 min and the supernatant pipetted into new eppendorf tubes. A volume of 50 µL of hydroxylamine hydrochloride (NH_3_OH.HCl) was added to the supernatant and the solution was agitated for 5 min. Additional volumes of 180 µL acetated buffer and 160 µL of 1, 10–phenanthroline were added to the supernatant and allow to stand for 30 min to form an intense orange-red complex of ferrous tris 1, 10–phenanthroline for absorbance measurement using UV-vis spectroscopy. The absorbance values recorded were converted into concentrations using a calibration curve plotted and the adsorption capacity of the CHAp adsorbent at time, $q_{t}$ in mg g^−1^ calculated for various concentrations using Equation (1):(1)$$q_{t}=\frac{C_{o}-C_{t}}{m}\times V$$
where $C_{o}$, $C_{t}$, denote the initial and final concentration of [Fe(H_2_O)_6_]^2+^/mg L^−1^ at time, *t*, respectively. *V* is the volume of solution, and *m* is the adsorbent dosage (g). A graph of $q_{t}$ (mg g^−1^) against *t* (min) was plotted to determine the various adsorption capacities at equilibrium, $q_{e}$ and subsequently calculate equilibrium concentration, $C_{e}$ using Equation (2):(2)$$C_{e}=C_{o}-\frac{q_{e}m}{V}$$

The $C_{e}$, $q_{e}$ and $q_{t}\;$ data obtained were used to determine the adsorption kinetics and maximum adsorption capacity, $q_{m}$ (mg g^−1^) of the CHAp adsorbent. The adsorption kinetics were explained using pseudo-first and–second-order Equations (5) and (6) whereas the $q_{m\;}$ modelled using nonlinear Langmuir and Freundlich Equations (3) and (4) whereas:(3)$$\ln\left(q_{e}-q_{t}\right)=\ln q_{e}-k_{1}t$$
(4)$$\frac{1}{q_{t}}=\frac{1}{k_{2}q_{e}^{2}}+\frac{1}{q_{t}}t$$
where $k_{1}$ and $k_{2}$ are the pseudo-first and -second order kinetics rate constants, respectively.
(5)$$q_{e}=\frac{q_{m}K_{L}C_{e}}{1+K_{L}C_{e}}$$
(6)$$q_{e}=K_{F}C_{e}^{1/n}$$
where $K_{L}\;$(L mg^−1^) and $K_{F}\;$(mg g^−1^) are Langmuir and Freundlich constant, respectively and *n* reveals the adsorption intensity.

### 2.4. Characterization Techniques

The CHAp adsorbent was characterized before and after adsorption studies using a Fourier-transform infrared spectrometer (FTIR Tensor 27, Bruker Corp., Billerica, MA, USA) over a wavenumber range of 400 to 4000 cm^−1^, and by Raman spectroscopy using a Technospex micro-Raman-532TEC-Ci spectrometer coupled with a µ-soft 2.0 analytical software. The CHAp was in powder form and analyzed with a single mode frequency-stabilized cobalt excitation laser (wavelength 532 nm and power 50 mW) at spectral range of 100–3400 cm^−1^ with 7 cm^−1^ resolution. Raman spectroscopic imaging was captured using 50× objective piece (Nikon, Tokyo, Japan, NA = 0.8). In morphological characterization, The CHAp image was acquired using a high-resolution FEI Helios Nanolab 650 FIB-SEM at the University of British Columbia (UBC) Centre for High-Throughput Phenogenomics. The electron image was acquired at a 1 kV accelerating voltage.

## 3. Results

### 3.1. SEM Image of CHAp

Figure 1 shows the scanning electron microscope (SEM) image of carbonate hydroxyapatite (CHAp). The morphology shows similar physical characteristics to bone hydroxyapatite [17]. The morphology has ellisodal, oval, spheriod, and rod-like structure [18,19] with no pore network.

### 3.2. IR Spectra Analysis of CHAp

Figure 2 shows infra-red (IR) spectra of pristine carbonate hydroxyapatite (CHAp) adsorbent (Figure 2a) and CHAp adsorbent after iron (II) adsorption from water (Figure 2b). The spectrum in Figure 3a show wavenumbers that are characteristic of functional groups of pristine CHAp. The wavenumbers that occurred at 1412, 871 and 713 cm^−1^ are carbonate ions whereas the wavenumbers at 1090 (ν3), 1029 (ν3), 963 (ν1), 600 (ν4), 563 (ν4) and 473 (ν2) cm^−1^ are symmetric vibrational modes of phosphate ions (PO_4_^3−^) in the crystal lattice of CHAp. In addition, the spectra show structural—hydroxyl ions (OH^−^) at wavenumbers 3574 and 634 cm^−1^. Figure 2b shows a stretching band of two water (H_2_O) molecules at 3395 and 1639 cm^−1^ which attaches itself to the carbonate ion (CO_3_^2−^) at wavenumbers 1412 cm^−1^ and thus reduces the transmission of the CO_3_^2−^ and the PO_4_^3−^ which indicates the introduction of iron (II) into the CHAp structure.

### 3.3. Raman Spectra Data Analysis and Spectroscopic Imaging of CHAp

Figure 3 shows the Raman spectra of the pristine CHAp adsorbent (Figure 3a) and after the adsorbent adsorption of iron (II) from water (Figure 3b). Figure 4a indicates PO_4_^3−^ (ν1) and CO_3_^2−^ (ν1)—B-type substitution vibrations of CHAp at wavenumber 962 and 1066 cm^−1^, respectively [20]. Three other CO_3_^2−^ vibrations are recorded at wavenumbers 1305, 1754, and 2433 cm^−1^ [21]. Figure 3b shows the introduction of four new spectra lines that occurs at wavenumbers 155, 287, 718, and 1090 cm^−1^ after the CHAp adsorbent [Fe(H_2_O)_6_]^2+^ adsorbate adsorption. Figure 3c,d shows spectroscopy images of pristine CHAp and residues after adsorbate adsorption, respectively. Figure 3c shows agglomerated grains of CHAp whereas Figure 3d displays a spherical phase of size 1.9–2.5 µm on the surface of the original CHAp.

### 3.4. Adsorption Test Results

Figure 4 shows the adsorption capacity, $q_{t}$ (mg g^−1^) of the CHAp adsorbent for varied contact times, *t* (min) after the removal of Fe (II) from the concentration of 20–100 mg L^−1^. The pictures in Figure 4f shows a change in colour from orange-red to the orange of 100 mg L^−1^ of Fe(II) solution which supports the fact that [Fe(H_2_O)_6_]^2+^ adsorption depended on contact time. The orange colour becomes pronounced after equilibrium time (120 min) where $q_{t}$ becomes $\;q_{e}$. The equilibrium adsorption capacity, $q_{e}\;$ of concentrations (20, 40, 60, 80, 100 mg L^−1^) at 120 min are 3.32, 6.63, 9.94, 13.24, 16.55 mg g^−1^, respectively.

#### 3.4.1. Adsorption Kinetics

Figure 5 show the linear plot of the adsorption kinetics of pseudo-first and -second order models, respectively. The pseudo-first-order assumes that the rate of adsorption of the adsorbate to time is directly proportional to the amount of available active sites on the adsorbent surface whereas the—second-order involves chemical groups that limits the adsorption rate through valance electron sharing between adsorbent and adsorbates [22]. The adsorption kinetics parameters determined from the intercept and slope of the model are shown in Table 1. The calculated equilibrium adsorption capacity, $q_{e}$ determine from the model were consistent to the experimental $q_{e}$ values in the pseudo-second-order model as indicated with the R^2^ values. The pseudo-second-order kinetic fits indicates that the aqua Fe(II) complex adsorption onto the CHAp is through valence electron sharing.

#### 3.4.2. Adsorption Isotherms

Figure 6 shows the nonlinear Langmuir and Freundlich isotherms model of the CHAp adsorbent material. Langmuir model predicted the maximum adsorption capacity, $q_{m}\;$(mg g^−1^) of the CHAp adsorbent whereas Freundlich explains the theoretical interaction mechanisms of the Fe(II) adsorbates and the CHAp adsorbent. Table 2 shows the model parameters where Langmuir and Freundlich show a close correlation to the experimental points with correlation coefficient, R^2^ value of 0.993 and 0.966, respectively. The Langmuir model predicted, $q_{m}\;$ of 45.87 mg g^−1^ and Freundlich gave a theoretical adsorption capacity, $K_{F}$ of 0.222 with 1/*n* value greater than 1. The Langmuir constant, $K_{L}$ value of 0.770 was used to determine the separation factor, $R_{L}$ of 0.061 from the Equation (7):(7)$$R_{L}=\frac{1}{1+K_{L}C_{o}}$$

The $R_{L}$ value determines the adsorption conditions. The adsorption is favourable when $0<R_{L}<1$, unfavourable when $R_{L}>$ 1, linear when $R_{L}=1$ and irreversible when $R_{L}=0$ [23].

## 4. Discussion

The IR and Raman spectroscopy data revealed B-type carbonate substitution in the CHAp adsorbent material: that is the presence of the structural hydroxyl group at wavenumber 3574 cm^−1^ in Figure 2a and the B-type CO_3_ ($\nu_{1}$) band at 1066 cm^−1^ in Figure 3a. Literature reports similar characteristics of the IR and the Raman observations [24]. The B-type carbonate substitution that occurs in CHAp was found to have an association with the aqua Fe(II) complex adsorption in Figure 2b and Figure 3b. The stretching H_2_O bands that appear at 3395 and 1639 cm^−1^ in Figure 2b are absent in Figure 2a which is an indication of the aqua Fe(II) complex adsorbate. The adsorbate attaches itself to the carbonate ion at wavenumber 1412 cm^−1^ to form iron(II) carbonate (FeCO_3_) reported in the literature as siderite [25]. The Raman spectrum in Figure 3b reveals unique spectra bands at 1090 cm^−1^ which bonds to the B-type CO_3_ ($\nu_{1}$) at 1066 cm^−1^ to further support siderite formation [20]. Carbonate (CO_3_^2−^) and phosphate (PO_4_^23−^) functional groups which are the main constituent of the CHAp have dipole bonds but because of the linear and tetrahedral shape these moities possess respectively, the dipole bonds are symmetric, and the net structure is nonpolar and possesses no dipole moment. It is strange that carbonate would be mainly responsible for the adsorbates adsorption as revealed by the IR and Raman spectra since the symmetric arrangements reduce the electronegativity of carbonate. On the contrary, the electronegativity of the CO_3_^2−^ in the CHAp(Ca_10−x/2_(PO4)_6_CO_3−x_OH_2_) structure increases in water and results in the attachments of the adsorbates. The reason is that the B-type carbonate substitution that occurs in the CHAp allows the structure to undergo stereoisomerism in water which permits the CO_3_^2−^ to conform to an asymmetric orientation and that introduces a dipole moment to increase electronegativity. The observation is reported in the authors’ earlier work [26] and Francesca Peccati et al. [27] also reports similar behaviour of the CO_3_^2−^ in A-type CHAp material. The reduction of IR spectrum bands in Figure 2b is attributed to the attachment of the adsorbates to the CO_3_^2−^ as confirmed by the Raman spectrum. The attachment increases the dipole moment and causes further stretching of the CO_3_^2−^ functional group which results in the reduction of CO_3_^2−^ band and the entire spectral band. Figure 3c,d show that there is a phase transformation from the original CHAp to a new possibly amorphous phase which is likely to be the Fe(II) substituted phase.

The IR and Raman spectra (See Figure 2 and Figure 3b) reveal that the aqua Fe(II) complex adsorption by CHAp was chemical adsorption (chemisorption) by sharing valence electrons between the carbonate functional groups and the aqua Fe(II) complex adsorbates. The SEM image of the CHAp shows nonporous morphology to support the chemisorption mechanism of the adsorption. The pseudo-second-order kinetic model confirms that functional groups are responsible for the adsorption but are time dependent. Equilibrium adsorption occurs at 120 min (See Figure 4)which indicates that the available CHAp active sites are occupied by large number of adsorbates at the saturation time, thus repelling further adsorption [22]. The chemisorption resulted in a high maximum adsorption capacity, $q_{m}\;$ of 45.87 mg g^−1^ and favourable adsorption as predicted by the non-linear Langmuir model (See Figure 6). The experimental values correlated well (R^2^ = 0.999) to the model because the CHAp adsorbent material is fairly hydrophobic and thus, not restricted to solubility at even very low concentrations under these conditions. The adsorption intensity (1/*n* value) predicted from the non-linear Freundlich model describes the extent of curvature the adsorption across the concentration range or the heterogeneity of the adsorbate site [23]. There are different types of the curvature of the adsorption isotherms namely, C-type, H-type, L-type, and S-type according Giles classification [28]. For C-type isotherms, the 1/*n* value is 1 and that means the relative adsorption (adsorption partition) of the adsorbates is the same through the entire concentration range tested. The L-type isotherm is the common curvature and for 1/*n* values from 0.7 to 1.0 indicate the dependency of the adsorbate concentration to relative adsorption. The adsorbate concentration has an inverse relationship to the relative adsorption. Thus, the adsorption site of the adsorbent material to the adsorbate is at saturation, and that reduces the adsorption capacity. The 1/*n* value of less than 0.7 defines highly curved isotherms (H-type) and is for adsorbates and adsorbent with highly affinity. For S-type isotherm which is most uncommon curvature has 1/*n* value greater than 1 and that represent adsorption at low concentrations with adsorbent material that have polar functional groups [29]. For our work, the non-linear Freundlich model predicted the CHAp adsorbent material to be S-type as indicated by 1/*n* value in Table 2. In addition, Table 2 show K_F_ value of 0.222 which indicates that the adsorption by the CHAp is extremely high based on the standard that K_F_ value of less than 0.5 or more than 50 is for extremely high or small adsorption, respectively.

The work here showing an adsorption capacity of 45.87 mg g^−1^ compares better with previous reports with lower adsorption capacities of HAp prepared from cow bone (2.39 [1]), and HAp modified with manganese oxide (0.606 mg g^−1^ [30]). The CHAp performs better than the HAp due to the carbonate substitution. The resulting carboxyl group produces an increased affinity for iron binding [31,32] whereas HAp only forms a colloidal Fe phosphate solid, which is attributed to the interaction of different phosphate species that combine with Fe (II) ions [33]. Also, compared to polymeric materials such as modified porous aromatic framework (PAF), the modified PAF (PAF-1-ET) showed a high adsorption capacity of 105 mg g^−1^ with an L-type isotherm, therefore rendering the material adsorption capacity incapable of meeting the standards for iron in drinking water due to the iron saturation in the absorbent during the absorption process [34]. This new CHAp adsorbent material could be used by environmental industries as a relatively cheap and ecologically benign method with which to purify drinking water without adverse effects.

## 5. Conclusions

The study prepared a carbonated hydroxyapatite (CHAp) adsorbent material from *Achatina achatina* snail shells and a phosphate-containing solution. The material was used to adsorb Fe (II) from a simulated aqua iron (II) complex water solution. Infrared and Raman spectroscopy data revealed that carbonate functional groups are mainly responsible for the aqua Fe (II) complex adsorption through stereoisomerism of the CHAp in water which resulted in asymmetric conformation of the carbonate groups. The spectroscopy data was supported by pseudo-second-order kinetics and the Freundlich model that indicated that the adsorption was chemisorption (S-type isotherm). Non-linear Langmuir model estimated the CHAp to have favourable adsorption with maximum adsorption capacity of 45.87 mg g^−1^ at room temperature and contact time of 180 min. The findings could be used in the design of novel CHAp bionano-based membranes materials that have no health implications after their use in remediation of iron (II) from drinking water.

## Figures and Tables

**Figure 1 materials-15-03253-f001:**
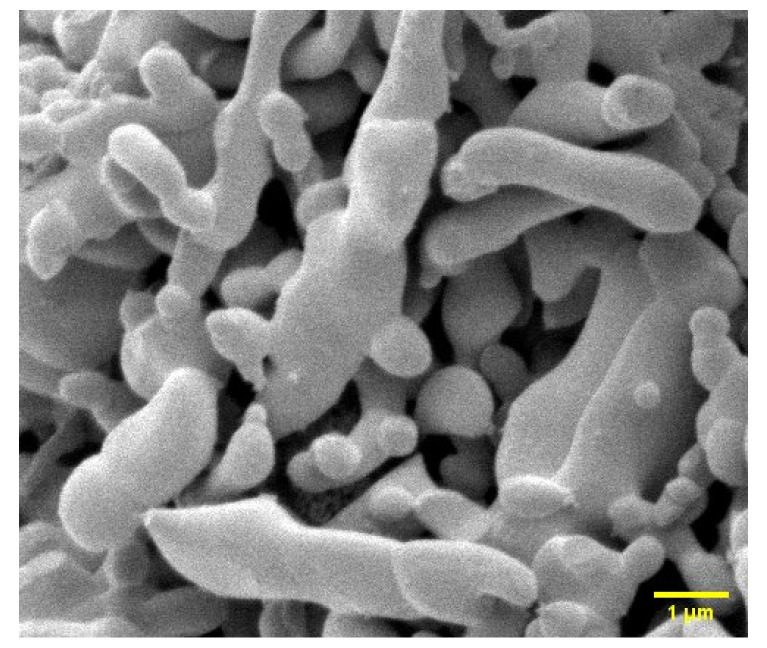
SEM image of CHAp.

**Figure 2 materials-15-03253-f002:**
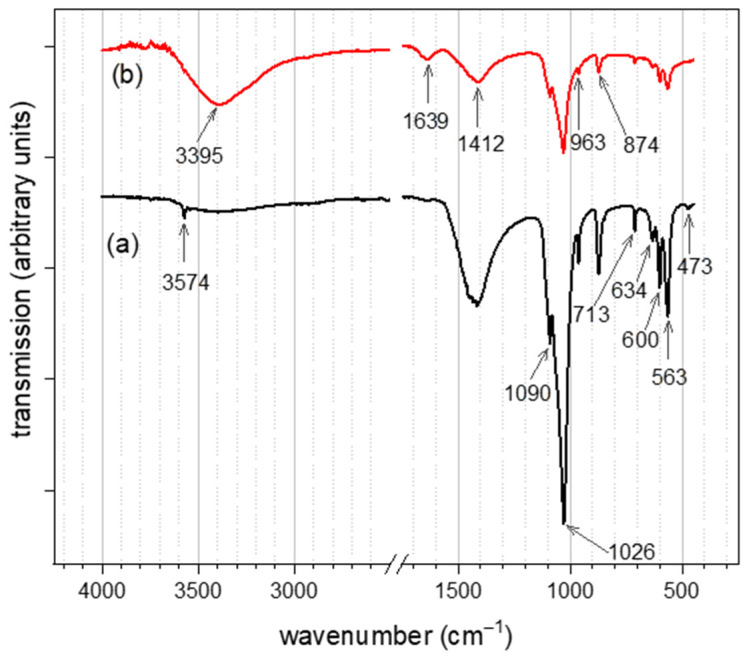
IR spectra of CHAp adsorbent material (**a**) pristine CHAp (**b**) residue CHAp after Fe(II) adsorption.

**Figure 3 materials-15-03253-f003:**
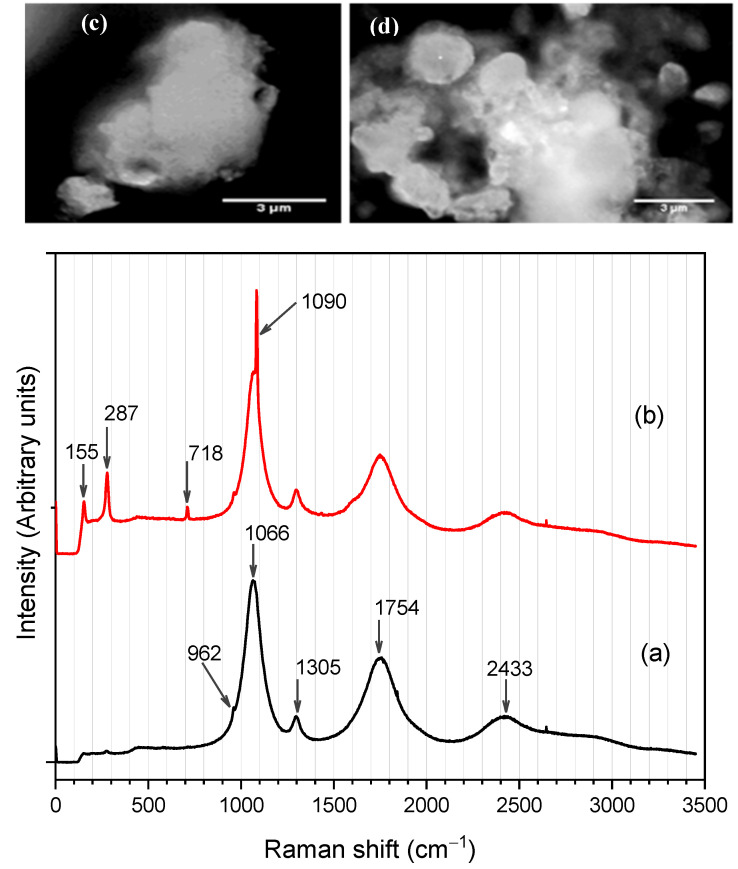
Raman spectra of CHAp adsorbent material (**a**) pristine CHAp (**b**) residue CHAp after Fe(II) adsorption (**c**,**d**) are Raman spectroscopic imaging of pristine CHAp and residue CHAp, respectively.

**Figure 4 materials-15-03253-f004:**
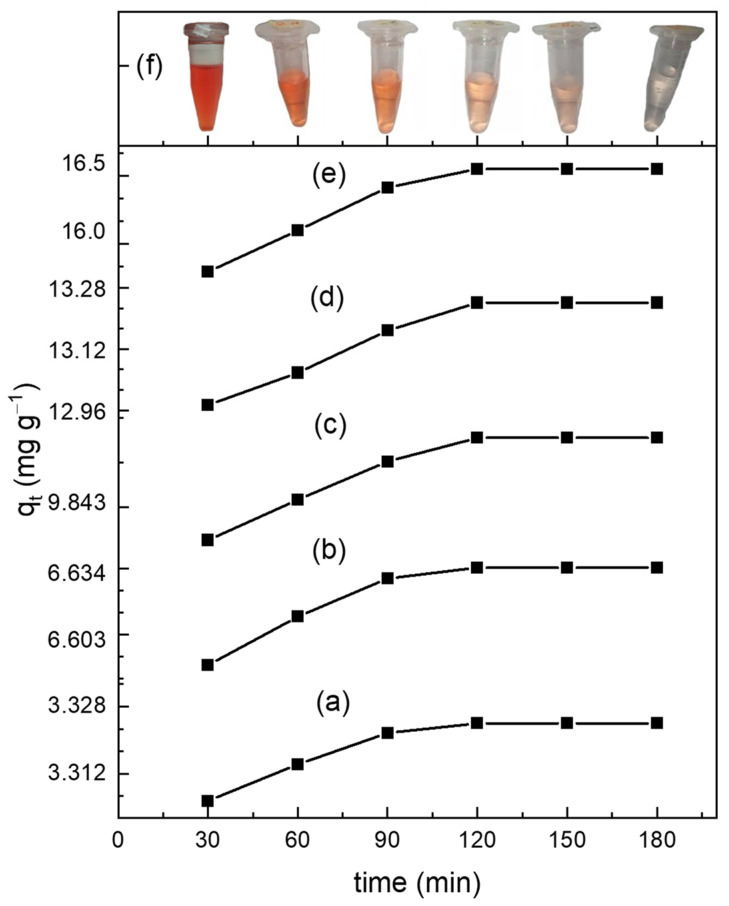
Show adsorption capacity at time, $q_{t}$ for various concentrations (**a**) 20 (**b**) 40 (**c**) 60 (**d**) 80 and (**e**) 100 mg L^−1^ of aqua Fe(II) complex adsorbates. The pictures shown in (**f**) from deep orange-red to light orange after the equilibrium adsorption time indicates the removal of the adsorbates from the Fe(II) solution. The pH of the before and after ChAp treatment is ~3.68 and 7.69, respectively.

**Figure 5 materials-15-03253-f005:**
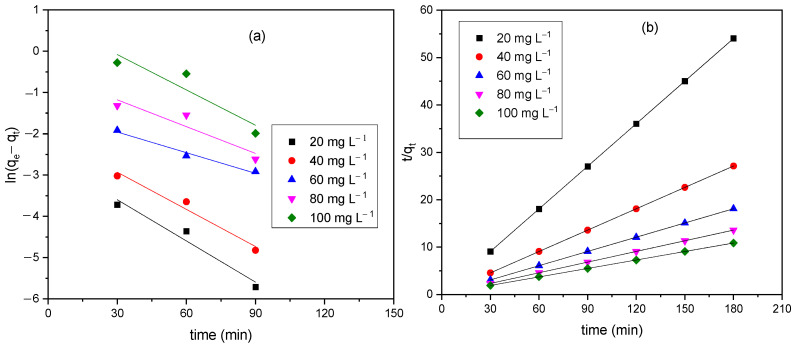
(**a**) pseudo-first-order (**b**) pseudo-second-order kinetic plots at room temperature for the CHAp adsorption of aqua Fe(II) complex adsorbates.

**Figure 6 materials-15-03253-f006:**
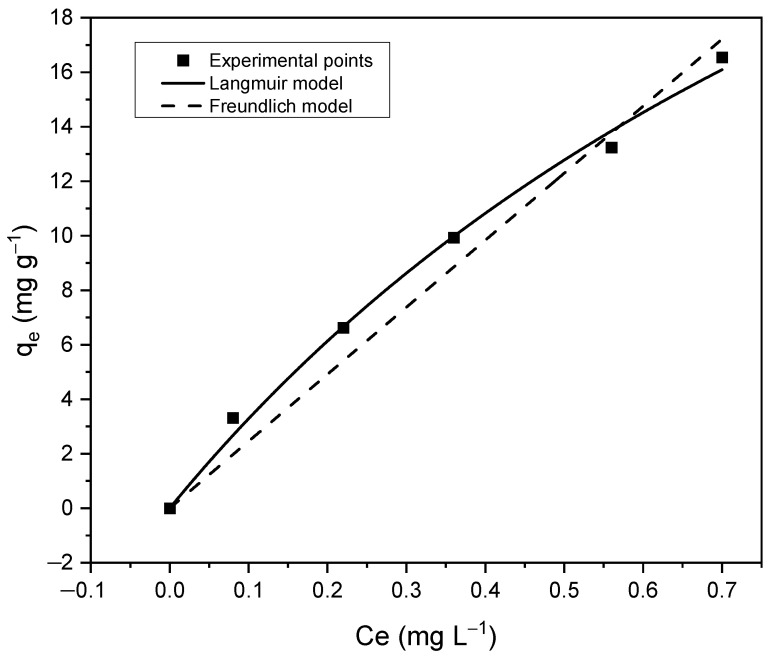
Nonlinear Langmuir and Freundlich isotherms at room temperature for the CHAp adsorption of aqua Fe(II) complex adsorbates.

**Table 1 materials-15-03253-t001:** Adsorption Kinetics parameters of Fe(II) onto CHAp adsorbent.

	Peudo-First-Order Model				Pseudo-Second-Order Model		
$\text{Conc}\;(\mathrm{mg}\;g^{-1})$	*q_e-exp_* ^a^ $\left(\mathrm{mg}\;g^{-1}\right)$	*q_e-cal_* ^b^ $\left(\mathrm{mg}\;g^{-1}\right)$	$K_{1}\;(\min^{-1})$ $\;\times \;10^{-4}$	R^2^	*q_e-cal_* ^b^ $\left(\mathrm{mg}\;g^{-1}\right)$	$K_{2}\;(g\;{\mathrm{mg}}^{-1}\;\min^{-1})$	R^2^
20	3.320	0.070	−1.850	0.920	3.320	1.074	1.000
40	6.630	0.130	−1.680	0.940	6.630	0.523	1.000
60	9.940	0.230	−0.927	0.962	9.980	0.152	0.999
80	13.240	0.590	−1.203	0.754	13.320	0.074	0.999
100	16.550	2.170	−1.587	0.728	16.780	0.027	0.999

^a^ and ^b^ indicates the experimental and theoretical *q_e_* values.

**Table 2 materials-15-03253-t002:** Langmuir and Freundlich adsorption parameters Fe(II) onto CHAp adsorbent.

Langmuir			Freundlich		
$q_{m}\;(\mathrm{mg}\;g^{-1}{)}_{\;}$	K_L_	R^2^	1/*n*	K_F_	R^2^
45.870	0.770	0.993	0.009	0.222	0.966

## Data Availability

Not applicable.

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
