# Peer review of "Snail Based Carbonated-Hydroxyapatite Material as Adsorbents for Water Iron (II)"

_materials, 2022, doi:10.3390/ma15093253_

Round 1

Reviewer 1 Report

This study has prepared a carbonated hydroxyapatite (CHAp) adsorbent material from Achatina achatina snail shells and a phosphate-containing solution. The adsorbent applied to adsorb water iron(II). Infrared and Raman spectroscopy were employed to reveal the mechanism of adsorption process.

  1. Why the author use this material to adsorb Fe(II) in solution? This should be explain in the Introduction part.
  2. The SEM images of CHAp should be provided.
  3. The correlation coefficient “r2” should expressed as “R2”,
  4. The writing problem should be carefully revised throughout the manuscript, including units “mgg-1” in P217 and “Table 2..” in P232.

Author Response

The authors express their gratitude to the reviewer for his valuable contribution. The following are the reviewer's comments and the author's response:

  1. Why the author use this material to adsorb Fe(II) in solution? This should be explain in the Introduction part.                                                                                                                                                                                   The reason has been inserted in lines 44–50.                                                                                                                                                      
  2. The SEM images of CHAp should be provided. 

    A SEM image of CHAp is included in Figure 1.

  3. The correlation coefficient “r2” should expressed as “R2”                                                                                                                                                   The correlation coefficient “r2” has been changed to “R2” in the writeup.
  1. The writing problem should be carefully revised throughout the manuscript, including units “mgg-1” in P217 and “Table 2..” in P232.

         “mgg-1” is changed to “mg g-1” in P217 and “Table 2.” in P232.

Reviewer 2 Report

This paper presents an interesting strategy searching for new adsorbent materials for water iron (II) based on Snail Based Carbonated-Hydroxyapatite. Although considerable works have been performed, several points must be improved for the acceptance of this manuscript.

  1. The authors did not provide sufficient background in the introduction to judge the novelty of their material.
  2. Please keep consistency in reference writing in the paper (ex. Page 2 line 48 [8][1] should replace with [1, 8]; Page 9 line 299 [22][23] should replace with [22, 23])
  3. Please use the same number of digits for the data presented on Table 1.
  4. The remove the descriptions after the each figure title (ex. Figure 1 remove: The wavenumbers at 3395, 1639 and 1412 cm-1 are iron carbonate (siderite) bands and that indicates that the pristine CHAp adsorbed the Fe(II) through the carbonate groups present in the CHAp. The reduction of the bands of the residue CHAp is due to the antisymmetric stretching of carbonate at 1412 cm-1, Figure 2: remove “The wavenumbers at 1090, 718, 287 and 155 cm-1 are iron carbonate (siderite) bands which complement the IR spectra data. ‘. If the information is not repeated, write it in the main text.
  5. Take care in using subscript and superscript (ex. Page 1 line 16 (Fe(H2O)6]2+); Page2 line 50 (Ca10-x/2(PO4)6CO3-xOH2)
  6. Why the pH was chosen at 7.69? I recommend choosing the optimal concentration and studying the pH influence on adsorption capacity.
  7. Could you explain why the temperature of room temperature was used?
  8. Please compare the adsorption capacities with the one presented in the research literature for similar adsorbents.
  9. How about the practical application value on an industrial scale?
  10. The conclusion section must be rewritten. The best optimal conditions, and best adsorbent capacity must be presented in this section.

Based on these, I advise the authors to rectify the above mentioned errors and I hope to re-evaluate the revised manuscript.

Author Response

The authors express their gratitude to the reviewer for his valuable contribution. The following are the reviewer's comments and the author's response:

1. The authors did not provide sufficient background in the introduction to judge the novelty of their material.

The reason has been inserted in lines 44–50 as also indicated by Reviewer1.

2. Please keep consistency in reference writing in the paper (ex. Page 2 line 48 [8][1] should replace with [1, 8]; Page 9 line 299 [22][23] should replace with [22, 23])

The inconsistencies in the references have been rectified.

3. Please use the same number of digits for the data presented on Table 1.

 In Table 1, the number of digits is changed to three decimal places.

4. The remove the descriptions after the each figure title (ex. Figure 1 remove: The wavenumbers at 3395, 1639 and 1412 cm-1 are iron carbonate (siderite) bands and that indicates that the pristine CHAp adsorbed the Fe(II) through the carbonate groups present in the CHAp. The reduction of the bands of the residue CHAp is due to the antisymmetric stretching of carbonate at 1412 cm-1, Figure 2: remove “The wavenumbers at 1090, 718, 287 and 155 cm-1 are iron carbonate (siderite) bands which complement the IR spectra data. ‘. If the information is not repeated, write it in the main text.

The figure descriptions are removed since they were in the write up.

5. Take care in using subscript and superscript (ex. Page 1 line 16 (Fe(H2O)6]2+); Page2 line 50 (Ca10-x/2(PO4)6CO3-xOH2)

This error is reverted.

6. Why the pH was chosen at 7.69? I recommend choosing the optimal concentration and studying the pH influence on adsorption capacity.

The recommendation is commendable; however, because our CHAp material is at a pH of around 8.12 and the Fe (II) water solution is also at a pH of around 3.68, which is now indicated in the write up, it sums up to a pH of 7.69. So, for physiological relevance, highly acidic and highly basic mediums will still need pH adjustment, which the paper tries to avoid.

7. Could you explain why the temperature of room temperature was used?

The ChAp is most stable at room temperature, and this reason is now included in the writeup in lines 74 and 75.

8. Please compare the adsorption capacities with the one presented in the research literature for similar adsorbents.

This is done in lines 276–278.

9. How about the practical application value on an industrial scale?

This is included in lines 286-288.

10. The conclusion section must be rewritten. The best optimal conditions, and best adsorbent capacity must be presented in this section.

The conclusion is modified and highlighted.

Round 2

Reviewer 1 Report

Accept in present form

Reviewer 2 Report

The author has made substantial improvements to this article. The manuscript can be accepted for publication in the present form.